# Regulation of host immunity by a novel *Legionella pneumophila* E3 ubiquitin ligase

**Shuxin Liu[1]\*, Chunlin He[1], Yong Zhang[1], Siyao Liu[1], Tao-Tao Chen[2], Chunxiuli Li[1], Dong Chen[1], Songying Ouyang[2], Lei Song[1], Jiaqi Fu[1]\*, Zhao-Qing Luo[1]\***

**1** Department of Respiratory Medicine, Center for Infectious Diseases and Pathogen Biology, Key Laboratory of Organ Regeneration and Transplantation of the Ministry of Education, State Key Laboratory for Diagnosis and Treatment of Severe Zoonotic Infectious Diseases, The First Hospital of Jilin University, Changchun, China, **2** Key Laboratory of Microbial Pathogenesis and Interventions-Fujian Province University, The Key Laboratory of Innate Immune Biology of Fujian Province, Biomedical Research Center of South China, Key Laboratory of OptoElectronic Science and Technology for Medicine of the Ministry of Education, College of Life Sciences, Fujian Normal University, Fuzhou, China

\* liushuxin@jlu.edu.cn (SL); fujiaqi@jlu.edu.cn (JF); luoz@jlu.edu.cn (Z-QL)

## Abstract

*Legionella pneumophila*, the causative agent of Legionnaires' disease, exists ubiquitously in natural and artificial water systems. This pathogen poses serious threat to human health. One salient feature of *L. pneumophila* pathogenesis is the hundreds of effectors delivered into host cells by its Dot/Icm transporter. These virulence factors interfere with multiple hosts signaling pathways to subvert host defense. The ubiquitin network is essential in host signaling involved in immunity and thus is a common target of *L. pneumophila* effectors. At least thirteen Dot/Icm effectors have been shown to function as E3 ubiquitin ligases that cooperate with the host ubiquitination machinery by distinct mechanisms. In addition, seven deubiquitinases (DUBs) have been characterized. Furthermore, effectors that utilize catalysis mechanisms that are chemically distinct from the canonical one found in eukaryotes have been reported, indicating that hijacking of the host ubiquitin network by *L. pneumophila* is extensive and complex. Here, we identified ubiquitin interacting proteins with a proximity labeling method and found that the effector Lug14 (Lpg1106) functions as a novel ubiquitin ligase. Lug14 works with the E2 UbcH5c to catalyze ubiquitination with a preference for $K_{11}$-linked chains by a mechanism that does not require a cysteine residue. Finally, we found that Lug14 targets ARIH2, a member of the host RBR E3 ligase family, leading to increased activation of the NLRP3 inflammasome in macrophages.

## Author summary

Successful pathogens have evolved virulence factors to co-opt the host ubiquitin network to counteract immunity or to redirect host cellular events to create a niche conductive for their replication. More than twenty of the approximate 330

**Data availability statement:** All relevant data are within the manuscript and its Supporting Information files.

**Funding:** This work was supported in part by the National Natural Science Foundation of China grants 32100137 (SL), 32370185 (JF), 32200141 (YZ), 32270185 and 32470179 (LS), 82225028 and 82172287 (SO), the National Key Research and Development Program of China 2021YFC2301403 (SO). The funders had no role in study design, data collection and analysis, decision to publish, or preparation of the manuscript.

**Competing interests:** The authors have declared that no competing interests exist.

effectors of the *L. pneumophila* Dot/Icm system have been shown to regulate ubiquitin signaling via various mechanisms. Here we tested the hypothesis that more effectors are involved in such regulation by using proximity labeling to identify *L. pneumophila* proteins potentially interacting with ubiquitin. Subsequent screening identified Lpg1106 (Lug14) as an E3 ubiquitin ligase. Biochemical analysis revealed that Lug14 coordinates with the E2 enzyme UbcH5c to catalyze the formation of multiple ubiquitin chain types, with a propensity for K-11 type. Lug14 targets the E3 ligase ARIH2, which in turn impact the activity of the NLRP3 inflammasome. In addition, the putative ubiquitin-interacting effectors identified in our study have laid the foundation for other assays relevant to ubiquitin signaling, which will further expand the boundary of the exploitation of the host ubiquitin network by *L. pneumophila*.

## Introduction

Ubiquitination is a post-translational modification that covalently attaches ubiquitin to target molecules through a reaction cascade mostly achieved via a reaction cascade catalyzed by three enzymes; this modification impacts virtually all cellular processes, particularly vesicle trafficking, development and immune response by altering the stability, activity or/and cellular location of the modified proteins [1]. As a result, co-option of ubiquitin signaling is a common strategy for pathogens to colonize hosts by elaborating virulence factors that function as ligase or deubiquitinase [2]. For example, pathogens in the *Salmonella*, *Shigella*, *Xanthomonas* and *Chlamydia* genus deliver effectors with E3 ligase or deubiquitinase activity by type III secretion systems to promote their survival and replication in hosts [3].

Among the bacterial pathogens known to hijack the ubiquitin network, *Legionella pneumophila* has emerged as an example that distinguishes itself in both the number of ubiquitin-relevant effectors and the richness of their mechanisms of action [4]. From the approximately 330 effectors delivered into host cells by *L. pneumophila* [5,6], at least 26 effectors have been found to co-opt the host ubiquitin network by diverse biochemical mechanisms [7,8]. Among them, at least 13 effectors have been shown to function as E3 ubiquitin ligases that work together with host E1 and E2 enzymes by distinct mechanisms. Some of these proteins possess structural domains such as F-box and U-box found in components of eukaryotic E3 ligase complexes, including LegU1, AnkB, LicA, PpgA, Lpg2525, LubX and GobX [9–15]. The cellular targets of some of these E3 ligases have been identified. For example, LegU1 catalyzes ubiquitination of the HLAB-associated transcript-3 (BAT3) to interfere with apoptosis, endoplasmic reticulum (ER) response and other processes important for cell homeostasis [11]. AnkB has been reported to induce host protein degradation by catalyzing K48-linked polyubiquitination, thus providing nutrients to the bacterium [16]. AnkB also interacts with the host protein Parvin B and modulates its ubiquitination [12]. SidC and SdcA attack Rab small GTPases such as Rab10 to

promote the biogenesis of the *Legionella*-containing vacuole (LCV) [17]. Lug15 ubiquitinates and recruits the t-SNARE Sec22b to the LCV [18]. RavN and Lpg2452 (a.k.a LegA14/SdcB) catalyzes ubiquitination by a mechanism akin to that of SidC, but their cellular targets remain elusive [19]. LubX functions as a metaeffector of SidH to downregulate its abundancy by ubiquitination and subsequent proteasome degradation [14], it also attacks Cdk2-like kinase 1, a host protein involved in cell division [13].

In agreement with the notion that the reversal of ubiquitination plays an important role in signaling [20], *L. pneumophila* has been found to code for a number of deubiquitinases (DUBs). Among these, the amino terminal portion of the members of the SidE family harbors a DUB domain with a canonical CHD catalytic triad of the CE clan of cysteine protease, these DUBs cleaves Lys11/48/63 chains with a preference for Lys63 [21]. LotA [22], Ceg23 (a.k.a. LotB) [23–25] and Lem27 (a.k.a. LotC) [25,26] belong to the ovarian tumor (OTU) family cysteine protease [27]; LotA is unique by harboring two active sites [22] and LotB is involved in controlling the association of the Sec22b with the LCV [28] whereas Lem27 appears to modulate ubiquitination of Rab10 induced by the bacterial E3 ligases SidC and SdcA [26]. RavD specifically cleaves linear ubiquitin chains to suppress the activation of immune response such as the NFκB pathway by host cells [29].

In addition to effectors that co-opt host ubiquitin signaling by mechanisms that require enzymes from the host's ubiquitination system, *L. pneumophila* also codes for effectors that catalyze ubiquitination by mechanisms that are a radical departure from the ATP-dependent E1-E2-E3 cascade. MavC catalyzes ubiquitination by transglutamination which crosslinks ubiquitin to Lys92 of UBE2N, an E2 enzyme mostly known to be involved in diverse cellular processes, particularly immunity [30]. This modification blocks the entry of activated Ub to its Cys87 active site [31,32], leading to the inhibition of the production of ubiquitin chains needed for the activation of immune factors such as NFκB [32,33]. The activity of MavC is counteracted by MvcA, which reverses the modification by removing ubiquitin from modified UBE2N [32,34]. Members of the SidE family proteins (SdeA, SdeB, SdeC and SidE) [35] catalyze phosphoribosyl ubiquitination by a two-step process involved in ubiquitin activation by ADP-ribosylation [36] and subsequent transferring of phosphoribosyl ubiquitin to serine residues on substrate proteins by a phosphodiesterase activity [37,38]. These enzymes target multiple structurally diverse proteins involved in cellular processes ranging from vesicle trafficking to autophagy and ER structure [25,29,36–38]. Serine ubiquitination of Rab33b by SidEs is required for the recruitment of Rab6A and consequently ER-resident SNARE proteins to the LCV [39]. How phosphoribosyl ubiquitination of proteins involved in autophagy benefits the bacterium awaits further investigation. The activity of SidEs is under stringent regulation by DupA and DupB, two effectors with phosphodiesterase activity that function to remove phosphoribosyl ubiquitin from modified proteins [40,41] and by SidJ, a calmodulin-dependent glutamylase that inhibits SidEs at later phases of *L. pneumophila* infection [42–46]. Finally, phosphoribosyl ubiquitin produced from modified substrates by DupA and DupB is reduced to native ubiquitin by sequential reactions catalyzed by LnaB and MavL, respectively [47,48]. These findings further highlight the importance of ubiquitin signaling in *L. pneumophila* virulence.

The discovery that a large cohort of Dot/Icm effectors can hijack the ubiquitin network indicates extensive co-option of this signaling mechanism by *L. pneumophila*. A number of these effectors harbor functional domains found in enzymes of eukaryotic origins, suggesting their acquisition by horizontal gene transfer. We consider the possibility that some effectors with activity relevant to ubiquitin signaling may be acquired by convergent evolution and thus have gone undetected in bioinformatics analysis. Because most proteins involved in ubiquitin signaling directly interact with the modifier, we explored Dot/Icm substrates that potentially modulate ubiquitination by the proximity labeling technology mediated by the promiscuous *Escherichia coli* biotin ligase mutant BirA* [49–51]. These efforts led to the identification of a collection of Dot/Icm substrates potentially interacting with ubiquitin, further experiments revealed that Lpg1106 has ubiquitin E3 ligase activity that targets a host E3 ubiquitin ligase to modulate immunity.

## Results

### Identification of *L. pneumophila* proteins that potentially interact with ubiquitin

Currently known E3 ubiquitin ligases in *L. pneumophila* were identified by bioinformatics and/or careful biochemical and structural analyses [11,13,15,19,33,36,46,52]. The existence of a large cohort of effectors involved in ubiquitin signaling suggests extensive exploitation of the host ubiquitin network by this bacterium, we thus explored the hypothesis that *L. pneumophila* codes for additional E3 ligases that are undetectable by methods that had been employed in previous studies. Because E3 ligases have the propensity to recognize ubiquitin by various mechanisms [53], we utilized the proximity labeling method based on the biotin ligase BirA* [49–51] to identify potential ubiquitin binding proteins. To this end, we expressed the Flag-BirA*-Ub fusion in *L. pneumophila* and enriched biotinylated proteins by beads coated with streptavidin and identified biotin-labeled proteins by mass spectrometric analysis [50] (Fig 1A and S1 Table). As expected, many effectors known to be involved in ubiquitin signaling were identified, indicating the effectiveness of our method (Fig 1B, **upper portion**). In addition, several Dot/Icm effectors not previously known to be involved in ubiquitin signaling were identified (Fig 1B, **lower portion**).

### The Dot/Icm substrate Lpg1106 has E3 ubiquitin ligase activity

Next, we examined the E3 ligase activity of the effectors identified in the screening by standard ubiquitination reactions containing the E2 UbcH5c using self-modification of the candidates as a readout. From the 26 proteins examined, we found that Lpg1106 displayed self-ubiquitination activity in an ATP-dependent manner, suggestive of an E3 ubiquitin ligase; it was thus designated as Lug14 (*Legionella* ubiquitin ligase gene 14) (Fig 2A). Under similar experimental conditions, self-ubiquitination also detectably occurred in reactions containing Lpg1751, Lpg1851 or Lpg0634 but the signals were considerably weaker (S1 Fig). These proteins were not pursued further in the current study. Further experiments with a standard dropout assay confirmed that self-modification occurred on Lug14 only in reactions containing all components required for ubiquitination to proceed (Fig 2B).

We also examined whether the level of self-ubiquitination by Lug14 increased when the reaction was allowed to proceed for longer durations. In reactions containing all components required for self-modification, ubiquitination became

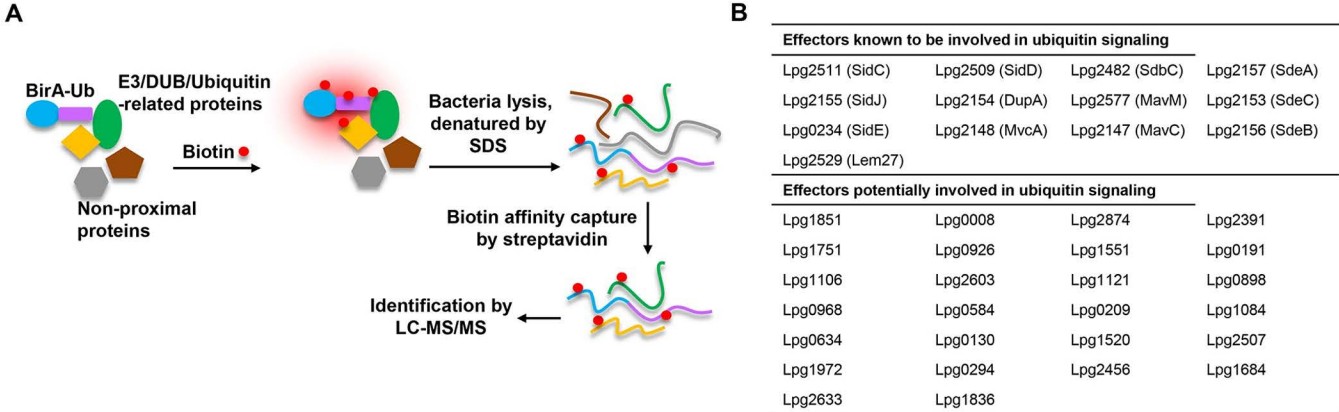

**Fig 1. Screening for ubiquitin-interacting Dot/Icm effectors.** A. Schematic representation of the experimental procedure for identification of *Legionella* proteins potentially interacting with ubiquitin. A BirA*-Ub fusion was expressed in the bacteria grown in medium supplemented with exogenous biotin. The shapes of different colors represent proteins potentially interacting with ubiquitin. Total protein of the bacteria was incubated with streptavidin beads and bound proteins were identified by mass spectrometric analysis. B. Dot/Icm effectors identified in our screenings. The proteins were presented in two groups, one (upper) includes effectors known to be involved in ubiquitin signaling and the second group (lower) includes effectors of unknown activity.

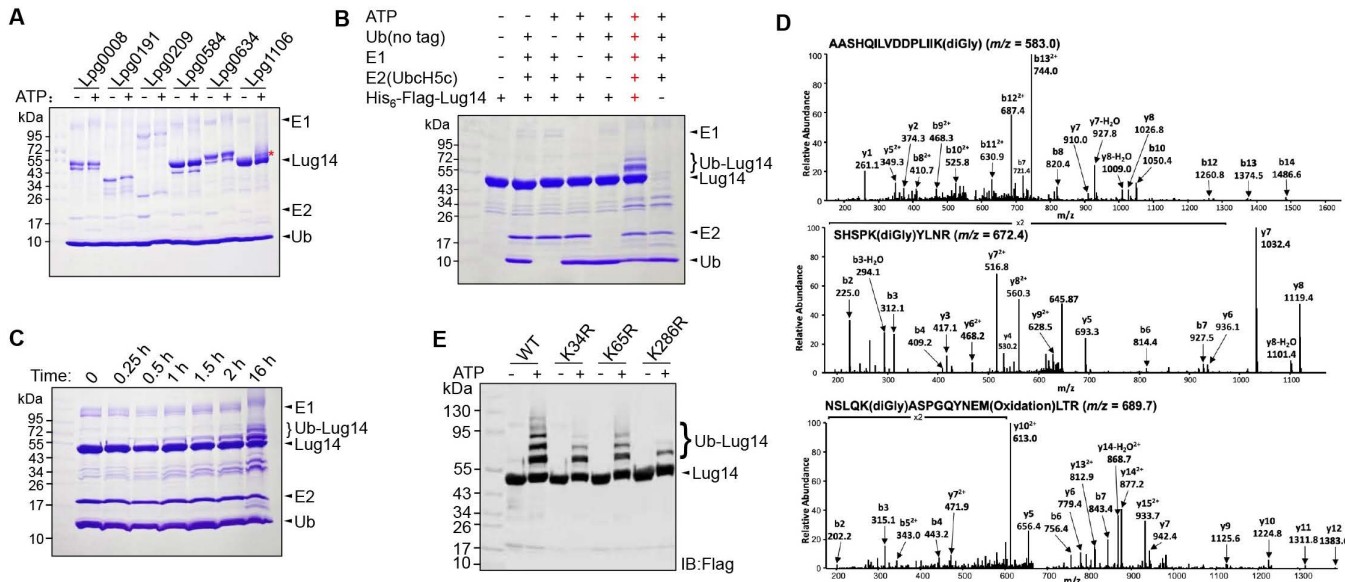

**Fig 2. The *Legionella* effector Lpg1106 (Lug14) has E3 ubiquitin ligase activity.** A. Lug14 displayed self-ubiquitination activity in biochemical assays. E1, UbcH5c, ubiquitin and His$_6$-tagged candidate proteins were incubated with or without ATP at 37°C for 16 h, proteins in the reactions were separated by SDS-PAGE and visualized by Coomassie brilliant blue (CBB) staining. Self-ubiquitination was determined by the production of protein species with MW higher than their native forms. B-C. Self-ubiquitination by Lug14 requires all reactants for canonical ubiquitination. Reactions containing ATP, ubiquitin, E1, UbcH5c and His$_6$-Flag-Lug14 were incubated at 37°C for 2 h (B). A series of reactions described in B were established and the reaction was terminated at the indicated time points prior to SDS-PAGE and detection by CBB staining (C). D-E. Identification of the modification sites on Lug14. Protein bands representing modified Lug14 were excised, digested with trypsin and analyzed by mass spectrometry. The spectra of K34 fragments, K65 and K286 were shown (D). Mutations in modified lysine residues of Lug14 reduced self-ubiquitination (E). Reactions containing Lug14 or its mutants were allowed to proceed for 2 h at 37°C prior to SDS-PAGE and detection by immunoblotting using Flag-specific antibody. In each case, similar results were obtained in at least three independent experiments.

detectable after 15 min incubation, and extension of the reaction time to 16 h did result in considerably more ubiquitinated Lug14 species (Fig 2C).

To further prove that self-ubiquitination indeed occurred on Lug14, we attempted to identify its lysine (K) residues that receive ubiquitin. To this end, the protein bands of Lug14 with molecular weight (MW) higher than that of native protein from biochemical reactions were excised and analyzed by mass spectrometry, which identified three tryptic fragments with an MW corresponding to the addition of the diGly remnant associated with canonical ubiquitination. Further LC-MS/MS analysis revealed that K34, K65 and K286 were the modified residues (Fig 2D). We then created mutants in which one of these three sites was substituted with arginine and tested each of them for self-ubiquitination. Although the level varied, each mutant exhibited reduced self-modification (Fig 2E), validating their role as ubiquitin recipient sites.

### Lug14 exhibits preference for the E2 conjugation enzyme UbcH5c

A canonical E3 ubiquitin ligase needs to cooperate with an E2 enzyme, which serves not only as a bridge for the transfer of activated ubiquitin from the E1 enzyme, but also a factor that often dictates such important features of ubiquitination as substrate selection and chain type of polyubiquitination [54]. To determine the E2 enzyme that is preferably utilized by Lug14, we examined its reactivity with several E2s commonly used by E3s of bacterial origin, including UbcH5a, UbcH5b, UbcH5c, UbcH6, UbcH7, UbcH8 and UbcH9. A series of reactions containing GST-Lug14 and each of these E2 enzymes were established and the reactivity was evaluated by the level of self-ubiquitination. Among these E2s tested,

self-modification by Lug14 occurred the most pronouncedly in reactions receiving UbcH5c (Fig 3A). We thus used UbcH5c in all ubiquitination reactions catalyzed by Lug14 in subsequent experiments.

### The E3 ubiquitin ligase activity of Lug14 does not require a cysteine residue

Among the three major families of E3s, RING E3s catalyze direct transfer of ubiquitin from the E2~ubiquitin conjugate to substrate, whereas HECT and RBR E3s utilize a catalytic cysteine that functions to receive activated ubiquitin from the E2~ubiquitin conjugate to form an E3~ubiquitin thioester intermediate from which ubiquitin is transferred to substrate [53]. Lug14 is a protein of 424 amino acids, with the exception of the region encompassing residues 131–301 predicted to contain ankyrin repeats potentially involved in protein-protein interactions, no other motif suggestive of known biochemical activity such as ubiquitination was detected by bioinformatics tools such as the HHpred algorithm [55] (Fig 3B), we nevertheless determined the potential involvement of a cysteine residue in the catalysis of Lug14 by constructing single, double and quadruple mutants (cysteine residues to alanine). All mutants retained the ligase activity at levels comparable to that

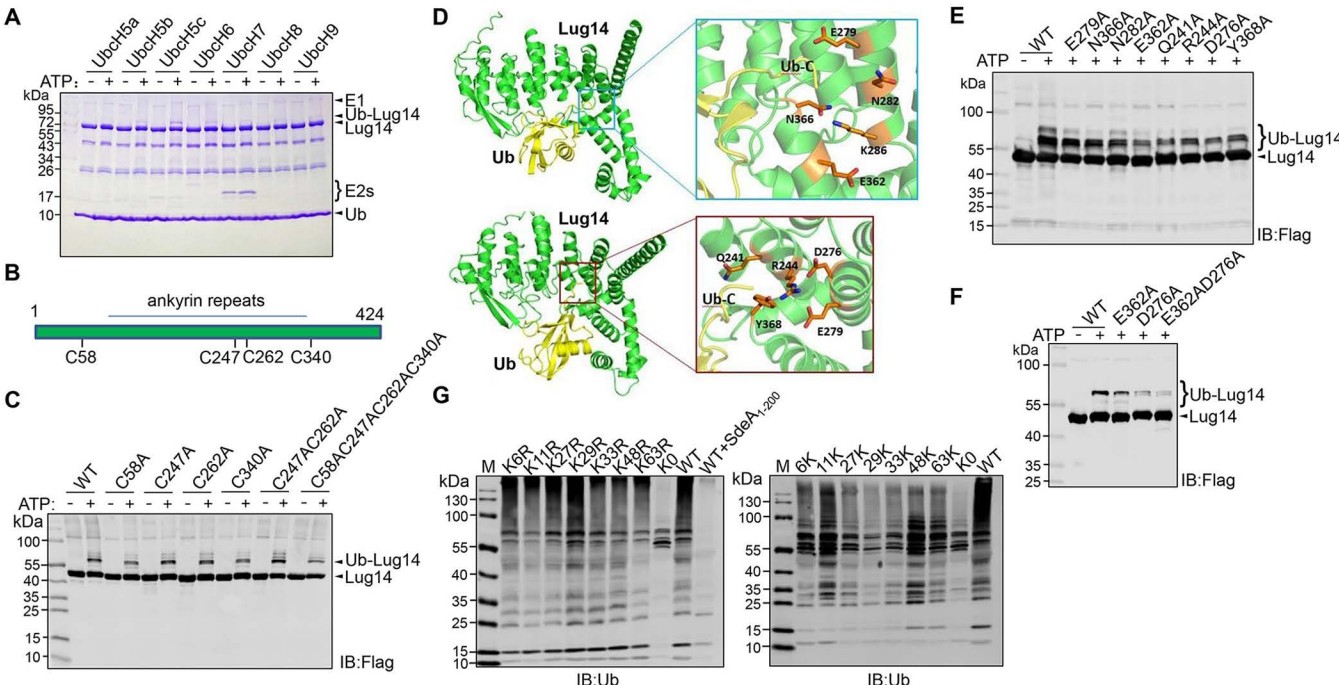

**Fig 3. Characteristics and property of Lug14.** A. UbcH5c is the preferred E2 enzyme for Lug14. A series of reactions each contained E1, ubiquitin, GST-Lug14 and the indicated E2 enzyme were allowed to proceed for 2 h at 37°C. Proteins separated by SDS-PAGE were detected by CBB staining. Results shown were one representative of three independent experiments with similar results. B-C. The E3 ligase activity of Lug14 does not need a cysteine residue. Recombinant Lug14 or its single, double and quadruple mutants (cysteine residues to alanine) was used in standard ubiquitination reactions containing E1, UbcH5c and ubiquitin. Reactions were allowed to proceed for 2 h at 37°C and samples resolved by SDS-PAGE were detected by CBB staining to determine ubiquitination by the formation of Lug14 species of higher MW. D-F. Construction of Lug14 substitution mutants defective in the E3 activity. The interface between Lug14 and ubiquitin predicted by AlphaFold and residues potentially involved in the engagement of these two proteins were identified (D). Lug14 mutants in which the residues predicted to be involved in the interactions had been mutated were used for ubiquitination assays (E). Mutations in modified lysine residues of Lug14 affected self-ubiquitination. Reactions containing Lug14 or its mutants were allowed to proceed for 2 h at 37°C prior to SDS-PAGE and detection by immunoblotting using Flag-specific antibody. In each case, similar results were obtained in at least three independent experiments. Note that the Lug14$_{E362AD276A}$ mutant has substantially lost the ability to catalyze self-ubiquitination (F). G. Lug14 catalyzes the formation of multiple types of ubiquitin chain types. Biochemical ubiquitination assays using E1, UbcH5c, Lug14 and a series of ubiquitin mutants that either carried only a single mutation in one of the lysine residues (left) or retaining only one lysine residue (right). K0 represents the ubiquitin mutant lacking all seven lysine residues. Reactions were allowed to proceed at 37°C for 2 h and proteins resolved by SDS-PAGE were detected by immunoblotting using Ub-specific antibody.

of wild-type protein (Fig 3C). These results suggest that Lug14 catalyzes the transfer of ubiquitin from charged UbcH5c by a mechanism that does not require a cysteine residue.

To obtain a substitution mutant of Lug14 defective in the ligase activity for subsequent experiments, we used AlphaFold [56] to predict the structure of the potential Lug14-Ub complex from which sites in Lug14 potentially involved in engaging ubiquitin were identified (Figs 3D and S2). Next, we created a total of 8 mutants each with a mutation in one of these sites. Biochemical ubiquitination assays revealed that only mutants $Lug14_{E362A}$ and $Lug14_{D276A}$ displayed a reduction in activity, and the activity of other mutants remained similar to that of wild-type protein (Fig 3E). We next constructed a mutant in which both E362 and D276 were replaced with alanine, the resulting mutant $Lug14_{E362D276}$ exhibited a more severe loss of activity than the single mutants (Fig 3F), suggesting that recognition of ubiquitin either in the UbcH5c-ubiquitin conjugate or ubiquitin being transferred during reaction by Lug14 is critical for its ligase activity.

### Lug14 catalyzes the formation of multiple types of ubiquitin chains

One key function of an E3 ligase is to work with its cognate E2 enzyme to control the type of ubiquitin chain added to its substrates [53], which sometimes provides insights into their roles in biological processes. We thus investigated the types of linkage formed by Lug14 by using a series of ubiquitin mutants that either carried only a single mutation in one of the lysine residues (Figs 3G, left and S3, left) or retaining only one lysine residue (Figs 3G, right and S3, right) along with the mutant $Ub_{K0}$ in which the all seven lysine residues have been replaced with arginine. The reactions were performed with UbcH5c, and ubiquitination was detected by immunoblotting. Except for the $Ub_{K0}$ mutant, formation of polyubiquitin chains was detected in reactions containing each of the lysine substitution mutants (Fig 3G). Interestingly, we observed more robust ubiquitination in reactions containing some of the ubiquitin mutants than wild-type ubiquitin, with $Ub_{K11}$ exhibiting the highest reactivity (Fig 3G). These results suggest that Lug14 preferably catalyzes the formation of the K11-type polyubiquitin chains.

### Expression of *lug14* is constant at different growth phases of *L. pneumophila*

To accommodate the need of effector activity at different phases of its interactions with host cells, the expression of many *L. pneumophila* Dot/Icm effectors is temporally regulated [57]. To assess the expression of *lug14*, we prepared Lug14-specific antibodies and used this reagent to examine its expression pattern by monitoring the level of Lug14 in bacterial cells grown at different phases in bacteriological medium. The protein level was relatively constant throughout the entire growth cycle, only a slight upregulation was observed when the bacteria entered the post-exponential phase (18 h) (S4 Fig). In contrast, the effector SidC is the most abundant in bacterial grown to the stationary phase [35] (S4 Fig) Thus, differing from many Dot/Icm substrates, *lug14* is only marginally induced in bacteria of the transmissive (virulent) state.

### Lug14 is dispensable for intracellular growth of *L. pneumophila* in common model hosts

To investigate the role of *Lug14* in intracellular bacterial replication, we infected mouse bone marrow-derived macrophages (BMDMs) from NOD-like receptor family CARD domain-containing protein 4 (NLRC4)$^{-/-}$ mice and *Dictyostelium discoideum* [58], respectively with relevant *L. pneumophila* strains, including Lp02(*dot/icm*±) [59], Lp03(*dotA*$^-$) [60], Lp02Δ*lug14* and Lp02Δ*lug14(*pLug14). The Lp02Δ*lug14* mutant grew indistinguishably to that of the wild-type strain in both hosts (Fig 4), indicating that similarly to the majority of characterized Dot/Icm substrates [7,61], deletion of *lug14* did not detectably affect intracellular replication of the bacterium in laboratory infection models.

### The host E3 ubiquitin ligase ARIH2 is a substrate of Lug14

To determine the role of Lug14 in *L. pneumophila* virulence, we sought to identify the host proteins targeted by this E3 ligase using the ubiquitin-activated interaction traps (UBAITs) method [62]. This strategy involves the construction of an

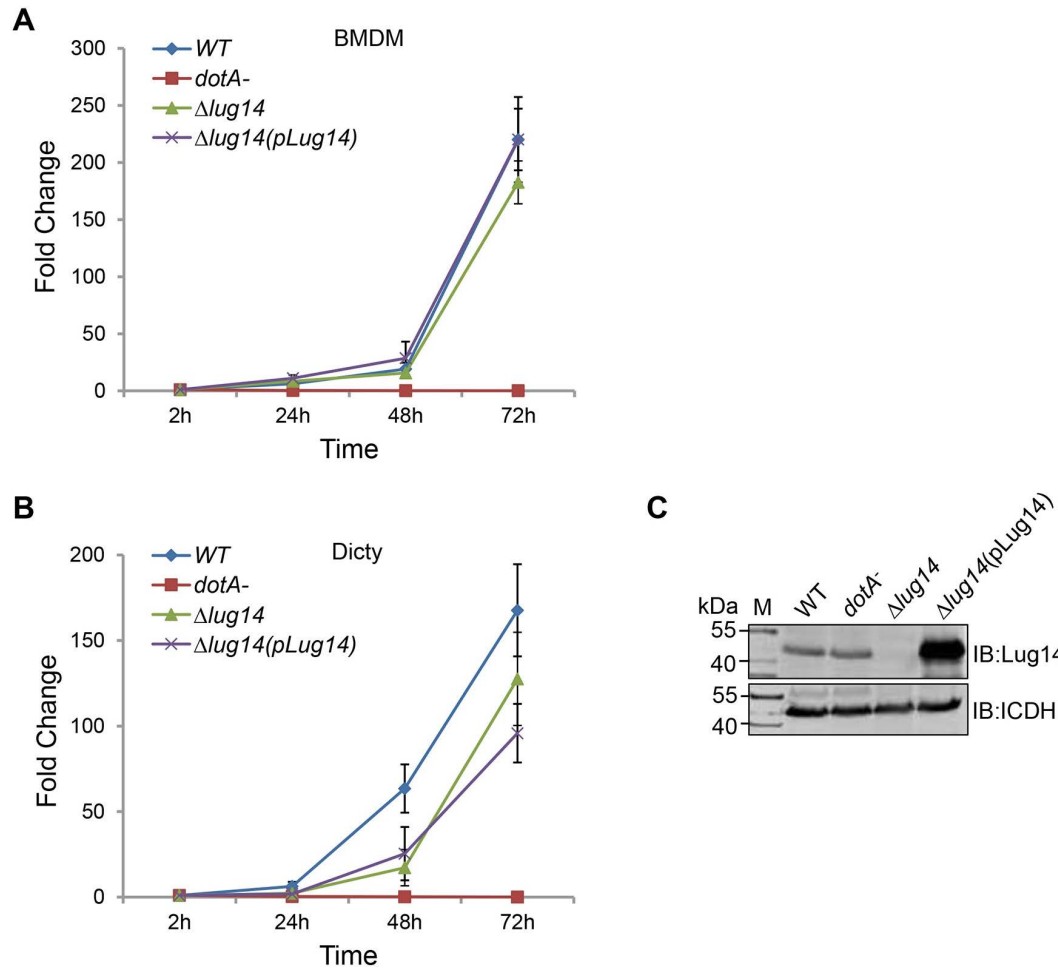

**Fig 4. Lug14 is not essential for optimal intracellular growth of *L. pneumophila*.** BMDMs from NLRC4$^{-/-}$ mice (A) or *D. discoideum* (B) were infected with the indicated *L. pneumophila* strains at an MOI of 0.05. Colony-forming-unit of the indicated times points were determined by spotting appropriately diluted lysates of infected cells onto bacteriological medium. Data shown are mean±s.e. from three samples of each strain. Similar results were obtained in three independent experiments. The expression of *lug14* in *L. pneumophila* strains is used for intracellular growth experiments. Equal amounts of bacterial cells were lysed and proteins resolved by SDS-PAGE were detected by immunoblotting with antibodies specific for Lug14, the metabolic enzyme isocitrate dehydrogenase (ICDH) was probed as a loading control (C).

E3-ubiquitin fusion that allows the C-terminus of ubiquitin to form an amide linkage to proteins potentially modified by the E3 ligase together with E1 and E2 enzymes, enabling co-purification of the E3 with candidate substrates. To this end, we established a cell line from HEK293T that stably expressed the GFP-Lug14-3xFlag-Ub fusion and used it to identify the target proteins of Lug14. A cell line that expressed a fusion containing RavN, another E3 ligase from *L. pneumophila* [19] was used as a control. Detection of the fusion protein with the Flag-specific antibody revealed the presence of proteins of MW higher than that of the predicted size of the fusion (Fig 5A). Subsequent mass spectrometric analysis identified several proteins that were more abundantly present in Lug14 samples, including HUWEI, an E3 ligase [63], MYO1D, an unconventional myosin that functions as actin-based motor protein with ATPase activity [64], CUL5, a core component of multiple Cullin-5-RING E3 ubiquitin ligase complexes [65] and the RBR family E3 ligase ARIH2, as potentially substrates for Lug14 [66] (S2 Table).

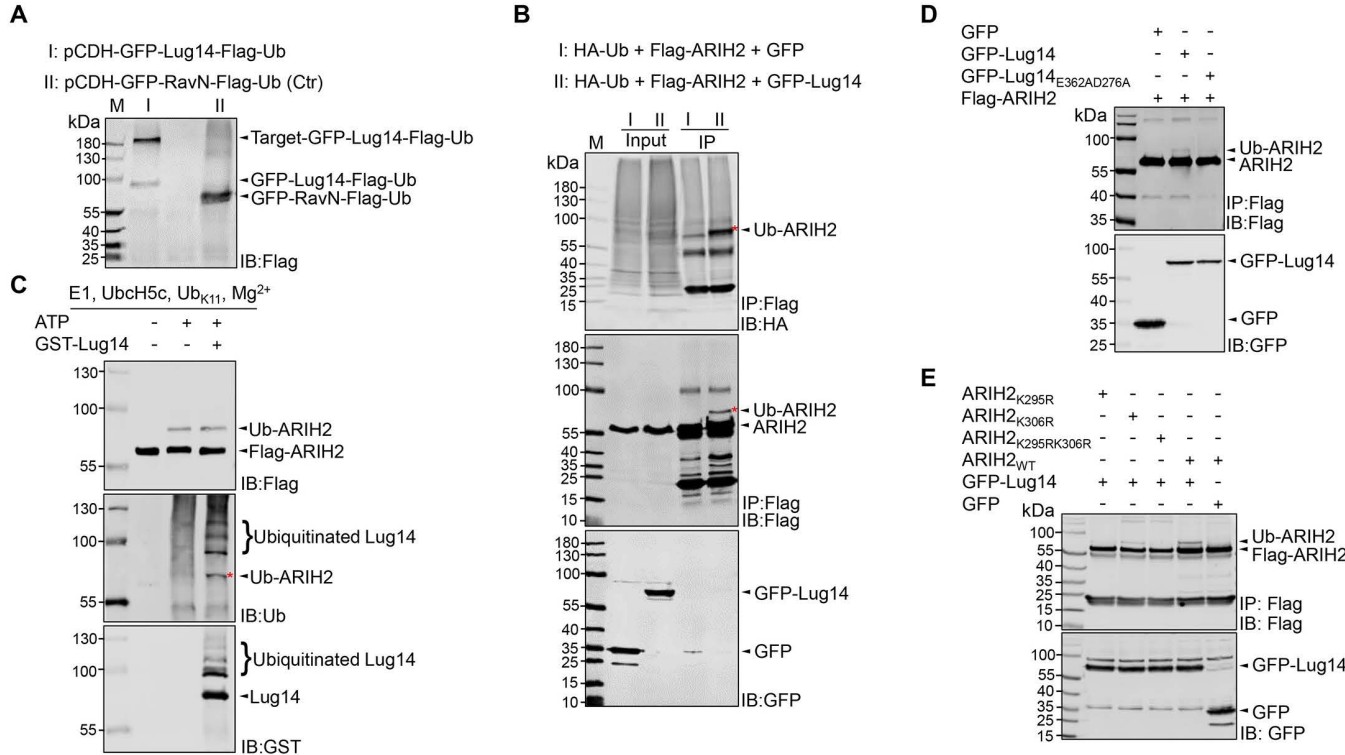

**Fig 5. The host E3 ubiquitin ligase ARIH2 is a substrate of Lug14.** A. Identification of the substrate of Lug14. Lysates of a cell line stably expressing the GFP-Lug14-3xFLAG-Ub fusion resolved by SDS-PAGE were detected by immunoblotting with the Flag-specific antibody. B. Lug14 catalyzes ubiquitination of ARIH2 *in cellulo*. HEK293T cells were co-transfected with plasmids encoding HA-ubiquitin, Flag-ARIH2 and GFP-Lug14. Samples receiving GFP were established as controls. After SDS-PAGE, samples were probed by immunoblotting with antibodies specific to HA, Flag and GFP, respectively. ARIH2 ubiquitination was determined by the formation of proteins of higher MW detected by antibody specific to HA or Flag. C. Lug14 catalyzes ubiquitination of ARIH2 in biochemical reactions. Purified Flag-ARIH2 was incubated with GST-Lug14 in reactions containing E1, UbcH5c (E2), Ub$_{K11}$, and ATP for 2 h at 37°C. After SDS-PAGE, samples were probed by immunoblotting with antibodies specific to Flag or Ub. ARIH2 ubiquitination was determined by the formation of proteins of higher MW detected by antibody specific to Flag or Ub. D. The Lug14$_{E362AD276A}$ mutant has lost the ability to catalyze ubiquitination on ARIH2. HEK293T cells were co-transfected with plasmids encoding Flag-ARIH2 and either GFP-Lug14 or GFP-Lug14$_{E362AD276A}$ for 24 h. Samples receiving GFP were established as controls. After SDS-PAGE, proteins were probed by immunoblotting with antibodies specific to Flag and GFP, respectively. ARIH2 ubiquitination was determined by the formation of proteins of higher MW detected by the Flag antibody. E. Lug14 catalyzes ubiquitination on K295 and K306 of ARIH2. HEK293T cells were co-transfected with plasmids encoding GFP-Lug14 and either wild type Flag-ARIH2 or each of the three mutants (Flag-ARIH2$_{K295R}$, Flag-ARIH2$_{K306R}$ and Flag-ARIH2$_{K295RK306R}$) for 24 h. Samples receiving GFP were established as controls. After SDS-PAGE, samples were probed by immunoblotting with antibodies specific to Flag and GFP, respectively. ARIH2 ubiquitination was determined by the formation of proteins of higher MW detected by the Flag antibody.

To test whether any of these candidate substrates is indeed ubiquitinated by Lug14, we first transfected HEK293T cells to express HA-Ub, GFP-Lug14 with Flag-tagged HUWEI, MYO1D, CUL5 or ARIH2. Whereas no ubiquitination signal was detected in samples expressing HUWEI, MYO1D or CUL5 (S5 Fig), co-expression of ARIH2 with GFP-Lug14 led to production of protein species of higher MW indicative of ubiquitination (Fig 5B). This result was confirmed by co-expressing Flag-ARIH2 with GFP or GFP-Lug14. Again, Lug14-catalyzed ubiquitination of ARIH2 was detected. In biochemical reactions, incubation of recombinant Flag-ARIH2 with Lug14 led to production of ubiquitinated ARIH2 (Fig 5C). To further investigate the ubiquitination of ARIH2 by Lug14, we constructed a plasmid expressing GFP-Lug14$_{E362AD276A}$, a mutant that has drastically reduced self-ubiquitination activity (Fig 3F). Whereas ubiquitination of ARIH2 by GFP-Lug14 was readily detected in HEK293T cells, modification by GFP-Lug14$_{E362AD276A}$ became undetectable (Fig 5D). Thus, Lug14 specifically catalyzes ubiquitination on ARIH2.

We next attempted to identify the ubiquitination sites on ARIH2 by subjecting the protein bands representing modified ARIH2 to mass spectrometric analysis, which revealed that K295 and K306 were the residues receiving ubiquitin (S6 Fig). To confirm the mass spectrometry results, we created mutants in which one of these two lysine residues was replaced with arginine and examined their suitability as Lug14 substrate. Our results showed that the K295R mutant displayed a more severe reduction in modification than the K306R mutant, suggesting that K295 is the primary modification site on ARIH2 in ubiquitination induced by Lug14. Furthermore, a mutant in which both K295 and K306 had been mutated to arginine had completely lost the ability to receive ubiquitin in reactions catalyzed by Lug14 (Fig 5E). Thus, Lug14 ubiquitinates ARIH2 at K295 and K306.

## Lug14 modulates the activation of the NLRP3 inflammasome

Because the modification sites of ARIH2 are close to its active site $Cys_{310}$ [66], we postulated that ubiquitination by Lug14 affects its E3 ligase activity. As a key regulator of inflammatory response, ARIH2 is important in defense against infection [67] and had been demonstrated to regulate NLRP3 activity in microphages by ubiquitination [68]. Thus, we tested whether Lug14-induced ARIH2 ubiquitination can inhibit its ubiquitination of NLRP3. Flag-NLRP3 and Flag-ARIH2 purified from transiently expressed HEK293T cells were incubated with recombinant GST-Lug14 in biochemical ubiquitination assays. We observed a significant increase in ARIH2 ubiquitination in sample receiving Lug14, which was accompanied by a decrease in NLRP3 ubiquitination. Of note, NLRP3 cannot be ubiquitinated by Lug14 (Fig 6A). These results indicate that the ubiquitination of ARIH2 by Lug14 led to inhibition of its E3 ligase activity.

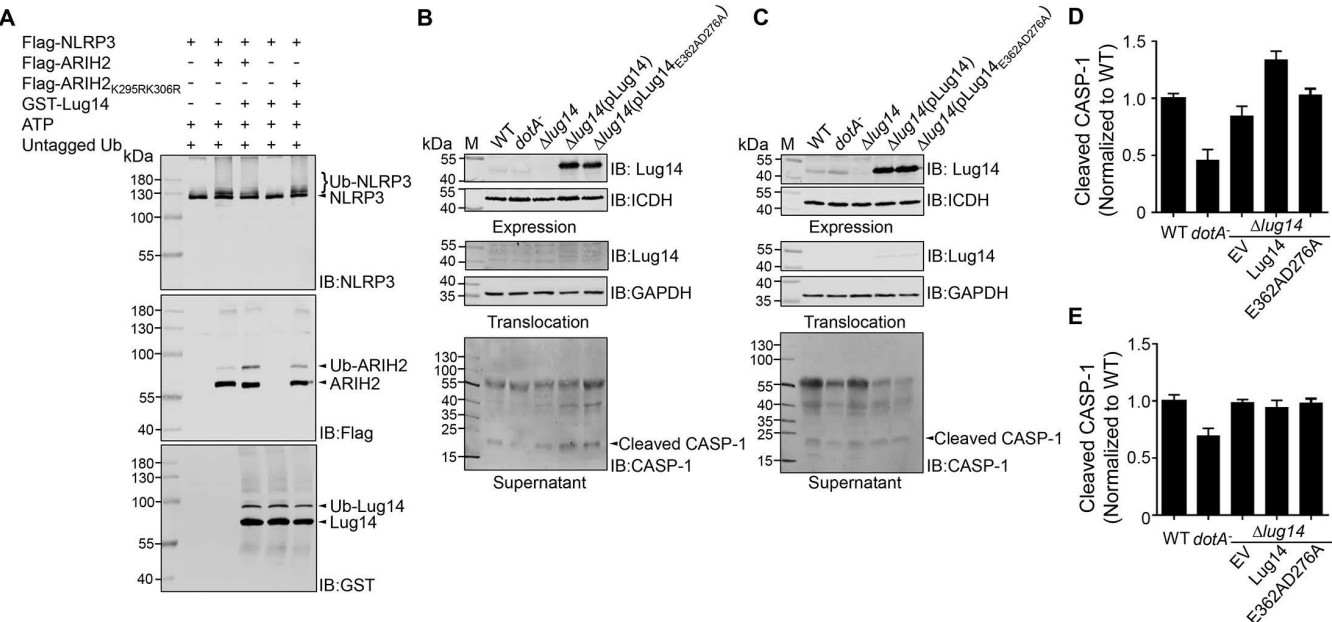

**Fig 6. Lug14 promotes NLRP3 inflammasome activation by inhibiting the E3 ligase activity of ARIH2.** A. Lug14-catalyzed ubiquitination inhibits the E3 ligase activity of ARIH2 toward NLRP3. Flag-NLRP3 and Flag-ARIH2 or Flag-ARIH2$_{K295RK306R}$ transiently expressed in HEK293T cells were precipitated with beads coated with the Flag antibody. Eluted proteins were used to determine the ligase activity of ARIH2 or ARIH2$_{K295RK306R}$ in reactions with or without Lug14. Ubiquitinated NLRP3, ARIH2 and Lug14 were detected by immunoblotting with anti- NLRP3, anti-Flag and anti-GST antibodies, respectively. B-E. Overexpression of Lug14 in *L. pneumophila* enhanced NLRP3 activation in infected macrophages. BMDMs from NLRC4$^{-/-}$ (B) and NLRC4$^{-/-}$NLRP3$^{-/-}$ (C) mice were infected with either Lp02(*dot/icm$^+$*), Lp03(*dotA$^-$*), Lp02Δ*lug14* or Lp02Δ*lug14(*pLug14) an MOI of 5 for 2 h. The levels of cleaved CASP-1 were assessed by immunoblotting. Quantification using ImageJ of the cleaved CASP-1 in infected BMDMs from NLRC4$^{-/-}$ and NLRC4$^{-/-}$NLRP3$^{-/-}$ mice was shown (D and E). Results shown were from three independent experiments.

Given that ARIH2 functions to negatively modulate the activation of the NLRP3 inflammasome, we asked whether Lug14 plays a role in NLRP3 activation in BMDMs infected by *L. pneumophila*. To this end, we infected BMDMs from NLRC4$^{-/-}$ mice with relevant *L. pneumophila* strains and examined the secretion of mature CASP-1 to measure NLRP3 activation (Fig 6B). Although there was no significant difference in the secretion of mature CASP-1 between samples infected with the wild-type and the Lp02Δ*lug14* mutant, infection with strain Lp02Δ*lug14(*pLug14) that overexpressed Lug14 led to more CASP-1 processing (Fig 6D). However, there was no significant difference in secretion of mature CASP-1 between *Lug14* gene knockout and complementation strains in BMDMs from NLRC4$^{-/-}$/NLRP3$^{-/-}$ mice (Fig 6C and 6E). These results suggest that ubiquitination of ARIH2 by Lug14 may create a condition more conductive for the activation of the NLRP3 inflammasome. We also examined the effects of NLRP3 on *L. pneumophila* virulence by comparing intracellular bacterial growth in BMDMs from NLRC4$^{-/-}$ and NLRC4$^{-/-}$/NLRP3$^{-/-}$ mice. Our results showed that *L. pneumophila* grew indistinguishably in BMDMs from these mouse lines (S7B and S7D Fig), suggesting that NLRP3 does not restrict in bacterial virulence under this experimental conditions.

## Discussion

*L. pneumophila* is exemplary in terms of the co-option of the host ubiquitin network by virulence factors. At least 26 Dot/Icm effectors have been shown to modulate ubiquitin signaling [8]. Our identification of Lug14 as an additional E3 ubiquitin ligase suggests even more extensive usage of effectors to exploit host cells by targeting ubiquitin signaling by *L. pneumophila*. Of note, albeit at relatively low levels, self-ubiquitination was also detectable for Lpg1751, Lpg1851 and Lpg0634 identified in our screenings (S1 Fig). The low ligase activity associated with these proteins in our assays may be caused by the fact that the E2s used were not optimal, or that the recombinant proteins were not properly folded. Alternatively, the signals may derive from reactions not catalyzed by a typical E3 ligase. More vigorous studies are needed to determine whether any of these proteins function to co-opt the ubiquitin network of host cells, and if so, what is the biochemical basis of their activity.

In contrast to E3s such as IpaH7.8 and IpaH7.8 of *Shigella* that target host proteins for degradation [69,70], ubiquitination by the majority of E3s from *L. pneumophila* does not appear to lead to protein degradation [8], the only exception is LubX, which functions as a metaeffector to promote SidH degradation via the ubiquitin-proteasome pathway [14]. Lug14 predominantly catalyzed the formation of K11-type ubiquitin chains (Fig 3G), which often impact the activity of the target protein without degradation [71]. In agreement with this notion, ubiquitination of ARIH2 inhibited its ligase activity (Fig 6A). Interestingly, SidC and SdcA, two E3 ligases from *L. pneumophila* also mainly catalyze the formation K11-type ubiquitin chains [52] on multiple small GTPases and SNARE proteins without causing protein degradation [72]. Whereas ubiquitination by Lug14 blocked the E3 ligase activity of ARIH2, modification by SidC and SdcA led to redistribution of small GTPases and SNARE proteins to the LCV [72]. Thus, how an E3 ligase impacts its targets depends upon not only the chain type but also the property of the targets.

Several E3 ligases from pathogens have been shown to target host E3s. For example, the *Salmonella* effector SopA [73] targets TRIM56 and TRIM65, two host E3 ligases involved in immunity [74]. Similarly, XopPXoo of the rice pathogen *Xanthomonas oryzae pv. oryzae* suppresses immunity by targeting the host E3 OsPUB44 [75]. That Lug14 targets the E3 ligase ARIH2 further highlights the importance of ubiquitin signaling in the intracellular life cycle of *L. pneumophila*.

The finding that Lug14 indirectly causes higher level activation of the NLRP3 inflammasome in cells infected with *L. pneumophila* appears counterintuitive as activation of immunity often is harmful to pathogen. Yet, it is well-established that modulation of host processes differently at distinct phases of infection is critical for *L. pneumophila* virulence. For example, the bacterium suppresses the NFκB branch of immunity within a few hours of entry into macrophages by effectors such as MavC [32], which is soon reversed by MvcA, which removes ubiquitin from UBE2N [34]. A few Dot/Icm effectors have been shown to activate NFκB, including LegK1, a kinase that phosphorylates IκB [76] and the phosphoryl AMPylase LnaB [47,48] whose mechanism of action in this regard remains elusive [77]. It is possible that Lug14 coordinates with

other Dot/Icm effectors to modulate the activity of NLRP3 at different stages of infection. Because an E3 ubiquitin ligase often targets multiple proteins to regulate vastly different biological processes [78], the inhibition of ARIH2 by Lug14 may regulate one or more cellular processes to make host cells more conductive to *L. pneumophila* replication. Future study aiming at other proteins regulated by ARIH2 may expand our understanding of the role Lug14 in *L. pneumophila* pathogenesis. Alternatively, NLRP3 has been shown to play roles in cellula processes not directly related to immunity [79,80], including type 1 diabetes [81], mitochondrial function [82] andendosomal transport [83], which are common targets of Dot/Icm effectors [84]. Lug14 may counteract the potentially negative effects of ARIH2 on NLRP3 to maintain a yet unrecognized cellular conditons benefitial to *L. pneumophila* infection.

Many bacterial E3 ligases utilize unique catalytic mechanisms for protein ubiquitination [85]. The fact that Lug14 does not display similarity to any E3 ligase, and it catalyzes ubiquitination independent of a cysteine residue suggests that its mechanism of action differs from those we currently understand. Such uniqueness offers valuable targets for the development of inhibitory molecules as such agents likely will have fewer side effects. Future biochemical and structural studies are needed to elucidate how this novel E3 ligase recognizes its substrate and how it catalyzes the transfer of ubiquitin. A better understanding of its mechanism of action will shed light into the study of E3 ligases that use similar mechanism to catalyze ubiquitination.

## Materials and methods

### Bacterial strains and plasmids

Bacterial strains, plasmids, and primers used in these studies are listed in S3 Table. *E coli* strains, *L. pneumophila* strains [59,60] and *D. discoideum* strain [58] used in this study were cultured as described in [26,52]. Gene deletion in *L. pneumophila* was carried out as described previously [46]. Restriction enzymes and T4 DNA ligase were purchased from NEB. Polymerase chain reaction (PCR) amplification was performed using TransStart Fast *Pfu* DNA Polymerase (AP221–03, TransGen, Beijing, China). Site-directed mutagenesis was performed by the Quikchange kit (Agilent) with primer pairs designed to introduce the desired mutations. All mutants were verified by double strand DNA sequencing.

### Protein expression and purification

*E. coli* strains for protein production were grown at 37°C in a shaker (250 rpm/min) to an $OD_{600nm}$ of 0.6~0.8. After adding IPTG (isopropyl thio-D-galactopyranoside) to a final concentration of 0.2 mM, bacteria were further cultured at 18°C for 16~18 h. Bacterial cells were then harvested by centrifugation at 4,000x*g* for 15 min. Harvested bacterial cells were resuspended in 30 mL PBS buffer (8 mM $NaH_2PO4$, 2 mM $KH_2PO4$, 136 mM NaCl, 2.6 mM KCl, pH 7.4) and lysed by a JN-Mini Low Temperature Ultrahigh Pressure Continuous Flow Cell Cracker (JNBIO, Guangzhou, China). Soluble fractions were collected by centrifugation at 12,000x*g* for 20 min at 4°C.

Lysates containing $His_6$-tagged proteins of interest were mixed with $Ni^{2+}$-NTA beads (Qiagen) for 1.5 h by rotation at 4°C, beads were washed extensively (at least 20x of the resin bed volume) with washing buffer (50 mM $NaH_2PO4$, 300 mM NaCl, 20 mM imidazole, pH 8.0). Proteins were eluted with 5 mL of elution buffer (50 mM $NaH_2PO_4$, 300 mM NaCl, 250 mM imidazole, pH 8.0). GST-tagged proteins were similarly purified using GST beads (Qiagen). Proteins were eluted with 5 mL of elution buffer (10 mM L-Glutathione reduced, 50 mM Tris-HCl, pH 8.0). Purified proteins were dialyzed overnight into 25 mM Tris-HCl (pH7.5), 150 mM NaCl and 10% (v/v) glycerol. Protein concentration was determined using the Bradford assay with BSA as the standard.

### Biochemical ubiquitination assays

Ubiquitination experiments were carried out in a reaction mixture (50 µL) containing 50 mM Tris-HCl (pH8.0), 5 mM $MgCl_2$, 5 mM ATP, 0.5 mM DTT, 100 nM E1, 100 nM UbcH5c(E2), 1 µM Ub and 0.5 µM purified Lug14. Unless otherwise stated,

reactions were allowed to proceed for 2 h at 37°C. Reactions were terminated with 5xSDS sample buffer. Samples were heated at 95°C for 5 min and analyzed by SDS-PAGE (15%-8% gradient gel) and analyzed by either Coomassie Brilliant blue staining or immunoblotting with antibodies against Ub and FLAG.

## Mass spectrometry analysis

To determine the specific ubiquitinated sites on target protein, the regions of ubiquitinated proteins were excised from gels staining with Coomassie Brilliant blue, digested in gel with trypsin, and the resulted peptides were dried on a Speed Vac vacuum concentrator and then re-suspended in an aqueous buffer before being subjected to LC-MS/MS analysis. The Orbitrap Eclipse mass spectrometer (Thermo Fisher Scientific) was used to identify and analyze the modified peptides.

## Cell cultures and transfection

HEK293T cells were grown in Dulbecco's modified minimum Eagle's medium (DMEM) supplemented with 10% fatal bovine serum (FBS). HEK293T cells were obtained from the ATCC and authenticated by short tandem repeat (STR) analysis and free of mycoplasma contamination as examined by a PCR-based test (Sigma, cat# MP0025). BMDMs were prepared from 6-week-old NLRC4$^{-/-}$ mice (Model Animal Research Center of Nanjing University, Nanjing, China) or NLRC4$^{-/-}$NLRP3$^{-/-}$ mice and cultured in RPMI 1640 medium supplemented with 20% L-cell conditioned medium (LCCM) and 20% FBS. The medium was supplemented with 100 μg/mL penicillin and 10 μg/mL streptomycin when necessary. All cell lines were grown at 37°C with 5% $CO_2$.

To establish a cell line stably expressing pCDH::*gfp-lug14-3xflag-ub*, we first package lentiviruses in HEK293T cells. Lentiviruses were generated by co-transfecting cells with 5 μg of shRNA-encoding plasmid pCDH::*gfp-lug14-3xflag-ub*, 3 μg psPAX2 and 2 μg pMD2.G plasmid using Lipofectamine 3000 (Invitrogen). Growth media was exchanged 6 h after transfection and lentiviruses-containing supernatant was harvested 48 h after transfection. Target cell lines were transduced with lentiviruses vectors-mediated inducible RNAi at a multiplicity of infection (MOI) of 5. Forty-eight h after transduction, medium containing 2 μg/mL puromycin was used to select for HEK293T cells harboring integrated pCDH::*gfp-lug14-3xflag-ub*. As a control, the same medium was added to untransduced cells seeded in a 24-well plate. The selective medium containing puromycin was replaced every 2–3 d, and the cells were visually inspected for toxicity. After one week, cell death began to occur in untransduced samples, the concentration of puromycin in medium for transduced samples was switched to 0.2 μg/mL and the cells were allowed to grow for 2 d. To isolate clones, cells diluted at a density of 1 cell per 100 μl were distributed in 96-well plates and wells that contained only 1 cell were identified under a fluorescence microscope. Several such clones were saved, and one was expanded into petri dishes and used for subsequent experiments.

## Bacterial infections and intracellular growth

For infection experiments, *L. pneumophila* strains were grown in AYE broth in a shaker (200 rpm) at 37°C to the post-exponential growth phase ($OD_{600}$ = 3.3–3.8) and bacterial motility was monitored under a light microscope. Intracellular growth was measured in NLRC4$^{-/-}$ BMDMs or *D. discoideum* cells. The indicated bacterial strains at an MOI of 0.05 were added directly to cells seeded in 24-well plates, extracellular bacteria were removed by washing 2 h after uptake. Then, the samples were refreshed with prewarmed RPMI 1640 medium containing 10% FBS and incubated at 37°C in a $CO_2$ incubator. At the indicated time points, samples were lysed with 0.02% saponin and incubated at 37°C for 30 min to allow complete cell lysis. The lysates were plated on CYE plates after proper dilutions. Bacterial colonies were counted after 4–5 d incubation at 37°C and counted to calculate the total CFUs in each sample.

BMDMs were seeded in 6-well plates and refreshed with prewarmed FBS- and antibiotic-free RPMI 1640 medium before infection. Then, the indicated bacterial strains were added directly to cells at an MOI of 5. For each infection, samples were centrifuged at 1,000x*g* for 5 min to promote sufficient contact of bacteria with host cells. Two hours after

infection, the culture supernatant was concentrated by the trichloroacetic acid (TCA) precipitation method and the cells were lysed with 0.02% saponin.

## Immunoprecipitation, antibodies, and immunoblotting

Transfected cells were resuspended with 1 ml NP40 lysis buffer supplemented with a complete protease inhibitor cocktail (sigma) for 10 min on ice, and the lysates were then centrifugated at 12,000x$g$ at 4°C for 10 min. Beads coated with Flag-specific antibody were added to cleared lysates and incubated on a rotatory shaker for 8 h at 4°C. After washing three times with 1 ml of cold NP40 lysis buffer, the beads were suspended in SDS loading buffer and were heated at 95°C for 5 min. After SDS-PAGE, proteins were transferred onto nitrocellulose membranes (Pall Life Sciences) for immunoblotting after being blocked in 5% nonfat milk in PBST buffer for 1 h. Primary antibodies used in this study and their dilutions are as follows: α-Flag (Sigma, cat# F1804, 1: 3000),α-Ub (Santa Cruz, cat# sc-8017, 1: 1000), α-GFP(Sigma, cat# G7781, 1:5000), α-GAPDH (Bioworld, cat#AP0063, 1:10000), α-ICDH (1: 20,000) [86], α-HA (Sigma, cat# H3663, 1:3000), α-NLRP3(Abcam, cat#ab263899, 1:3000), α-CASP-1 (Santa Cruz, cat# sc-514, 1: 1000). Antibodies specific to Lug14 were generated by immunization of rabbits with purified His$_6$-Lug14 using a standard procedure and were used at 1:1000. Washed membranes were incubated with appropriate IRDye secondary antibodies and signals were detected and analyzed by an Odyssey CLx system (LI-COR).

## Supporting information

**S1 Fig. Screening of proteins obtained in the TurboID for E3 ligase activity.** A. Assessment of E3 ligase activity of the 26 candidates using self-modification as a readout. E1, UbcH5c, ubiquitin and GST-tagged effectors were incubated with or without ATP at 37°C for 16 h. Proteins were separated by SDS-PAGE and visualized by CBB staining. B. Lpg1751, Lpg1851 and Lpg0634 displayed weak but detectable self-ubiquitination activity in biochemical assays. After SDS-PAGE, proteins were visualized by CBB staining or immunoblotting with antibodies specific to GST. Self-ubiquitination was determined by the production of protein species with MW higher than their native forms.
(TIF)

**S2 Fig. Distribution of prediction errors for ranges of AlphaFold prediction confidence.** A. The per-residue local Distance Difference Test (IDDT) scores for the predicted Lug14-Ub complex models. B. The PAE metrics for the predicted Lug14-Ub complex models show the predicted relative position error for each residue sequence, with low-confidence values in red and high-confidence values in blue.
(TIF)

**S3 Fig. The amounts of each Ub mutant added in each reaction were equal.** Biochemical ubiquitination assays using E1, UbcH5c, Lug14 and a series of ubiquitin mutants that either carried only a single mutation in one of the lysine residues (left) or retaining only one lysine residue (right). K0 represents the ubiquitin mutant lacking all seven lysine residues. All reactants were added into test tubes in ice, half of the sample in each reaction were taken to determine the amount of Ub detected by Coomassie brilliant blue staining.
(TIF)

**S4 Fig. The expression of *lug14* only slightly increases at the post exponential phase.** Saturated cultures of *L. pneumophila* strain Lp02 were diluted in AYE medium to OD$_{600}$ of 0.1 and the subcultures were grown at 37°C for 24 h. Bacterial growth was determined by measuring OD$_{600}$ (A) and samples withdrawn at intervals of 3 h were probed for Lug14 by immunoblotting (B). Lysates of the Lp02Δ*lug14* mutant grown to post exponential phase were included as a control. ICDH was probed as a loading control.
(TIF)

**S5 Fig. Lug14 did not detectably catalyze ubiquitination on HUWEI, MYO1D or CUL5.** HEK293T cells were co-transfected with plasmids encoding HA-ubiquitin, GFP-Lug14, Flag-tagged HUWEI, MYO1D or CUL5. Samples receiving GFP were established as controls. After SDS-PAGE, samples were probed by immunoblotting with antibodies specific to Flag and GFP, respectively.
(TIF)

**S6 Fig. Identification of the recipient lysine residues on ubiquitinated ARIH2 induced by Lug14.** Protein bands representing ubiquitinated ARIH2 were excised and digested with trypsin. The peptides characteristic of ubiquitination bearing the diGly remnant were detected and the modified lysine residues were mapped by LC/MS analysis. The spectra of fragments harboring K295 and K306 were shown.
(TIF)

**S7 Fig. NLRP3 does not restrict intracellular growth of *L. pneumophila*.** BMDMs from NLRC4$^{-/-}$ (A) or NLRC4$^{-/-}$/NLRP3$^{-/-}$ (B) mice were infected with the indicated *L. pneumophila* strains at an MOI of 0.05. Colony-forming-unit of the indicated times points were determined by spotting appropriately diluted lysates of infected cells onto bacteriological medium. Data shown are mean±s.e. from three samples of each strain. Similar results were obtained in three independent experiments. The expression of *lug14* in *L. pneumophila* strains is used for intracellular growth experiments. Equal amounts of bacterial cells were lysed and proteins resolved by SDS-PAGE were detected by immunoblotting with antibodies specific for Lug14, the metabolic enzyme isocitrate dehydrogenase (ICDH) was probed as a loading control (C). The wild-type strain grew indistinguishably in both hosts (D).
(TIF)

**S1 Table. *L. pneumophila* proteins that potentially interact with ubiquitin identified by the proximity labelling screen.**
(DOCX)

**S2 Table. Candidate targets of Lug14 identified by UBAITs.**
(DOCX)

**S3 Table. Bacterial strains, plasmids and primers used in the study.**
(DOCX)

## Acknowledgments

Mass spectrometric analysis was performed in the core facility of the First Hospital of Jilin University.

## Author contributions

**Conceptualization:** Shuxin Liu, Zhao-Qing Luo.

**Data curation:** Shuxin Liu, Tao-Tao Chen, Jiaqi Fu, Zhao-Qing Luo.

**Formal analysis:** Shuxin Liu, Tao-Tao Chen, Jiaqi Fu, Zhao-Qing Luo.

**Funding acquisition:** Shuxin Liu, Yong Zhang, Songying Ouyang, Lei Song, Jiaqi Fu.

**Investigation:** Shuxin Liu, Chunlin He, Yong Zhang, Siyao Liu, Tao-Tao Chen, Chunxiuli Li, Dong Chen, Jiaqi Fu.

**Methodology:** Shuxin Liu, Tao-Tao Chen, Jiaqi Fu, Zhao-Qing Luo.

**Project administration:** Zhao-Qing Luo.

**Resources:** Songying Ouyang, Lei Song, Jiaqi Fu, Zhao-Qing Luo.

**Software:** Tao-Tao Chen, Jiaqi Fu.

**Supervision:** Zhao-Qing Luo.

**Validation:** Shuxin Liu, Chunlin He, Yong Zhang, Siyao Liu, Tao-Tao Chen, Chunxiuli Li, Dong Chen, Jiaqi Fu.

**Visualization:** Shuxin Liu, Tao-Tao Chen, Jiaqi Fu, Zhao-Qing Luo.

**Writing – original draft:** Shuxin Liu.

**Writing – review & editing:** Jiaqi Fu, Zhao-Qing Luo.

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
