## [Decision Letter · Decision Letter 0]

28 Mar 2025

Regulation of host immunity by a novel Legionella pneumophila E3 ubiquitin ligase

PLOS Pathogens

Dear Dr. Luo,

Thank you for submitting your manuscript to PLOS Pathogens. After careful consideration, we feel that it has merit but does not fully meet PLOS Pathogens's publication criteria as it currently stands. Therefore, we invite you to submit a revised version of the manuscript that addresses the points raised during the review process.

Please submit your revised manuscript within 60 days May 27 2025 11:59PM. If you will need more time than this to complete your revisions, please reply to this message or contact the journal office at plospathogens@plos.org. Please include the following items when submitting your revised manuscript:

We look forward to receiving your revised manuscript.

Kind regards,

Dario S. Zamboni, Ph.D.

Academic Editor

PLOS Pathogens

Thomas Guillard

Section Editor

PLOS Pathogens

Editor-in-Chief

PLOS Pathogens

orcid.org/0000-0003-2946-9497

Editor-in-Chief

PLOS Pathogens

orcid.org/0000-0002-7699-2064

**Additional Editor Comments:**

The concerns raised by Reviewer #3 regarding the mechanistic details of Lug14 E3 ligase activity are relevant, but more importantly, the key issues raised by Reviewer #2 should be carefully considered, particularly the possibility that ARIH2 may have substrates other than NLRP3. In this context, the manuscript would be significantly strengthened by the inclusion of data demonstrating additional ARIH2 substrates. This point is especially relevant, as the rationale for*Legionella* enhancing NLRP3 activity does not align well with the current understanding of how this bacterium interacts with host immunity. As noted, *Legionella* evolved in protozoan hosts that lack the NLRP3 inflammasome, and in mammalian cells, inflammasome activation typically restricts bacterial replication. Therefore, the idea that Legionella would evolve to promote NLRP3 activation should be further substantiated.

**Journal Requirements:**

At this stage, the following Authors/Authors require contributions: Shuxin Liu, Chunlin He, Yong Zhang, Siyao Liu, Tao-Tao Chen, Chunxiuli Li, Dong Chen, Songying Ouyang, Lei Song, Jiaqi Fu, and Zhao-Qing Luo. Please ensure that the full contributions of each author are acknowledged in the "Add/Edit/Remove Authors" section of our submission form.

4) We notice that your supplementary Figures, and Tables are included in the manuscript file. Please remove them and upload them with the file type 'Supporting Information'. Please ensure that each Supporting Information file has a legend listed in the manuscript after the references list.

5) Please ensure that the funders and grant numbers match between the Financial Disclosure field and the Funding Information tab in your submission form. Note that the funders must be provided in the same order in both places as well.

**Reviewers' Comments:**

Reviewer's Responses to Questions

**Part I - Summary**

Reviewer #1: The manuscript by Shuxin Liu et al utilized proximity labeling technology combined with mass spectrometry to identify effectors of Legionella pneumophila that may affect the ubiquitin signaling pathway. Further in vitro auto-ubiquitination assays confirmed the E3 ubiquitin ligase activity of Lug14 (Lpg1106) and revealed that Lug14 primarily cooperated with the E2 UbcH5c, using a mechanism independent of cysteine residues to predominantly catalyze the formation of K11-linked ubiquitin chains. Further assays identified ARIH2 as the host target of Lug14. ARIH2 is a member of the host RBR E3 ubiquitin ligase family and is capable of ubiquitinating NLRP3 to negatively regulate the activation of the NLRP3 inflammasome. Lug14-mediated ubiquitination of ARIH2 attenuated the ubiquitination of NLRP3 and enhanced the activation of the NLRP3 inflammasome during Legionella pneumophila infection.

This study is very interesting. The discovery of Lug14 strengthens our understanding of how Legionella pneumophila effectors regulate host ubiquitination signaling pathways to interfere with immune signaling. The whole manuscript was well prepared. It should be accepted for publication after minor revesion.

Reviewer #2: In this work, Lui at al report on a new E3 ligase - Lug14 - from the human pathogen L. pneumophila that they propose modulates the NLRP3 inflammasome in infected macrophages. Legionella encodes a large number of functionally diverse UBLs. The authors identify Lug14 using a clever proximity ligation screen for ubiquitin (Ub)-binding proteins produced by the bacterium. The authors provide mechanistic insight into the enzymatic function by identifying a Ub binding pocked, through Alphafold modeling, and a putative substrate ARIH2, through a proteomics screen. ARIH2 is a E3 Ub ligase that negatively regulates the NLRP3 inflammasome (among other substrates), thus the authors propose that Lug14 enhances NLRP3 activity in infected cells by targeting ARIH2 for degradation. Overall, the data are consistent with Lug14 being a UBL, however based on the data and the experimental strategy I question whether ARIH2 is the physiologically relevant target of Lug14 which significantly temper my enthusiasm for the study. Conceptually, Legionella enhancing NLRP3 activity through degradation of AREH2 makes zero sense for a number of reasons: (1) the natural host of the bacterium, unicellular protozoa, lack the NRPL3 inflammasome and (2) inflammasome activation is restrictive for Legionella replication thus Lug14 activity, according to the author’s model, should interfere with bacterial replication.

Reviewer #3: This manuscript by Liu et al. describes the identification and characterization of a novel E3 ubiquitin ligase, Lug14 (Lpg1106), from Legionella pneumophila. The authors employ a proximity labeling approach to identify ubiquitin-interacting proteins, leading to the discovery of Lug14, which functions by catalyzing ubiquitination with a preference for K11-linked chains. The study reveals that Lug14 targets the host E3 ligase ARIH2 and modulates NLRP3 inflammasome activation in macrophages. This work adds to our understanding of how L. pneumophila manipulates the host ubiquitin network to influence host cell processes during infection.

**Part II – Major Issues: Key Experiments Required for Acceptance**

Reviewer #1: None.

Reviewer #2: (I) Lug14 capacity to ubiquitinate ARIH2 in a classical in vitro ubiquitination assay (Figs 5C, D and E) where all the components are in excess, which normally results in a robust substrate modification, is not impressive. Based on the data I might argue that monoubiquitinated ARIH2 is the most likely product produced by Lug14 in those reactions. Based on the data in Figs 5C, D and E, I would estimate that < 5% of ARIH2 is ubiquitinated. This is a problem because unlike in those assays, Lug14 under physiological conditions when translocated in the host cytosol by the bacterium is likely present at a very low amounts (as most Legionella effectors). The concern is that ARIH2 might not be a physiologically relevant target for Lug14. The paucity of data from NLPR3 and ARIH2 ubiquitination or functionality assay under infection conditions is not reassuring.

(II) Data in Fig 6A shows that ARIH2 ubiquitinates NLRP3 despite Lug14 addition (high molecular smear above NLRP3). Although, the data is very difficult to interpret because the authors decided to append a Flag tag to both ARIH2 as well as NLRP3 and did not include sufficient experimental control conditions to determine what the single band they designate as Ub-NLRP3 actually represents. A good control in this experiment would be to include the ARIH2 K295R K306R mutant protein, which could not be ubiquitinated by Lug14. I should point out that there are ARIH2 substrates other than NLRP3 in the literature that the authors have not explored (PMIDs 22037423, 23213454, 24076655, 23179078), including Cullin-based UBLs and ikBb. Thus, it is possible that they are missing the correct substrate (especially because of lack of NLRP3 functional data – see point III)

(III) The data demonstrating Lug14 translocated by the bacteria under infection conditions ubiquitinating ARIH2 and enhancing NLRP3 activity is lacking (Fig. 6B-C notwithstanding). The functional data in Fig 6B-C (presumably showing consequences for the NLRP3-inflammasome) are not rigorous. There are a large number of well-established molecular tools for assessment of NLRP3 activity thus a detailed analysis showing the consequences of Lug14 translocation in macrophages during infection specifically on NLRP3 functionality is needed in this work. This is critical point because depending on the situation multiple distinct inflammasomes can be triggered by Legionella in macrophages.

Reviewer #3: Mechanistic details of Lug14's E3 ligase activity: The authors conclude that Lug14 does not require a cysteine residue for its ligase activity, suggesting a non-canonical mechanism. However, the precise catalytic mechanism remains unclear. The authors identified E362 and D276 as important residues, but additional structural or biochemical evidence would strengthen the proposed non-canonical mechanism.

**Part III – Minor Issues: Editorial and Data Presentation Modifications**

Reviewer #1: I have the following minor concerns.

(1) In Figure 3D & F, it will be better to test the mutational effects of the E362 and D276 residues on the interaction between Lug14 and Ub.

(2) In Figure 3G, does the K11R mutation in Ub reduce the Lug14's auto-ubiquitination level, relative to wild-type Ub?

(3) In Figure 5, does Lug14 catalyze the K11-linked polyUb on ARIH2 in vitro?

(4) In Figure 6B, it will be better to include the infection result of the deletion strain complemented with the catalytically inactive Lug14 and its effects on NLRP3 inflammasome activation.

Reviewer #2: (I) The authors state that Lug14 does not conjugate Ub to a Cysteine residue for substrate transfer (a proposed novelty in enzyme functionality) because individual single Cys point mutations did not abrogate Ub acceptance. However, two of the residues C247 and C262 are sufficiently proximal that might be functionally redundant. A double mutant should be used to eliminate that possibility.

(II) From the data in Fig 3G the authors postulate that Lug14 has preference for K11 Ub chains, however given the differences in Lug14 amounts in each reaction and the lack of quantitative analysis it is hard to draw that conclusion. It is unclear if equal amounts of each Ub mutant is added in each reaction (the immunoblot for Ub is missing). I am curious, can the authors comment on why WT Ub is the least efficiently incorporated in Lug14 in those assays?

(III) The mass spec data from the proximity labelling screen should be provided as supplementary data source (including total AND unique spectral counts for each protein). In the same vein, the data in S1 table should also include unique spectral counts not just total spectral counts.

(IV) I would recommend that the intracellular growth assays in Fig 4 be presented as fold change in CFUs rather that total CFUs for easier data interpretation.

(V) Based on the OD data curve it seems to me that only log phase bacteria are sampled in S2 figure. I don’t see clear stationary phase. Also, the authors can include a control such as FlaA, which varies based on the growth phase.

(VI) For the benefit of non-specialist readers of this manuscript, it would be informative for the authors to include a short description of the UBAIT methodology they used at around line 303.

Reviewer #3: Figure 3G: The authors conclude that Lug14 prefers K11-linked chains, but the difference between K11 and some other chain types (e.g., K6, K33) appears modest. Statistical analysis would strengthen this conclusion.

Line 247-259: The authors used AlphaFold to predict the structure of the Lug14-Ub complex. Given the importance of this prediction for identifying potential interaction sites, further details on the model quality (pLDDT scores, PAE values) would be valuable.

PLOS authors have the option to publish the peer review history of their article (what does this mean? ). If published, this will include your full peer review and any attached files.

**Do you want your identity to be public for this peer review?** For information about this choice, including consent withdrawal, please see our Privacy Policy .

Reviewer #1: No

Reviewer #2: No

Reviewer #3: No

**Figure resubmission:**

**Reproducibility:**



---

## [Editor Report · Decision Letter 1]

8 Sep 2025

Dear Professor Luo,

We are pleased to inform you that your manuscript 'Regulation of host immunity by a novel Legionella pneumophila E3 ubiquitin ligase' has been provisionally accepted for publication in PLOS Pathogens.

Best regards,

Dario S. Zamboni, Ph.D.

Academic Editor

PLOS Pathogens

Thomas Guillard

Section Editor

PLOS Pathogens

Sumita Bhaduri-McIntosh

Editor-in-Chief

PLOS Pathogens

orcid.org/0000-0003-2946-9497

Michael Malim

Editor-in-Chief

PLOS Pathogens

orcid.org/0000-0002-7699-2064

The authors have appropriately addressed the reviewers’ concerns by clarifying the role of Lug14-mediated ubiquitination of ARIH2, expanding the discussion on NLRP3 modulation beyond canonical inflammasome activity, and highlighting the significance of identifying a novel E3 ligase and its substrate. This is a very nice and thorough piece of work, and the manuscript is now suitable for acceptance.
---

## [Editor Report · Acceptance letter]

Dear Professor Luo,

We are delighted to inform you that your manuscript, " 

Regulation of host immunity by a novel Legionella pneumophila E3 ubiquitin ligase," has been formally accepted for publication in PLOS Pathogens.

Best regards,

Sumita Bhaduri-McIntosh

Editor-in-Chief

PLOS Pathogens

orcid.org/0000-0003-2946-9497

Michael Malim

Editor-in-Chief

PLOS Pathogens

orcid.org/0000-0002-7699-2064